# State-dependent brainstem ensemble dynamics and their interactions with hippocampus across sleep states

Tomomi Tsunematsu[1,2,3,4], Amisha A Patel[1], Arno Onken[5], Shuzo Sakata[1]*

[1]Strathclyde Institute of Pharmacy and Biomedical Sciences, University of Strathclyde, Glasgow, United Kingdom; [2]Super-Network Brain Physiology, Graduate School of Life Sciences, Tohoku University, Sendai, Japan; [3]Frontier Research Institute for Interdisciplinary Sciences, Tohoku University, Sendai, Japan; [4]Precursory Research for Embryonic Science and Technology, Japan Science and Technology Agency, Kawaguchi, Japan; [5]School of Informatics, University of Edinburgh, Edinburgh, United Kingdom

**Abstract** The brainstem plays a crucial role in sleep-wake regulation. However, the ensemble dynamics underlying sleep regulation remain poorly understood. Here, we show slow, state-predictive brainstem ensemble dynamics and state-dependent interactions between the brainstem and the hippocampus in mice. On a timescale of seconds to minutes, brainstem populations can predict pupil dilation and vigilance states and exhibit longer prediction power than hippocampal CA1 neurons. On a timescale of sub-seconds, pontine waves (P-waves) are accompanied by synchronous firing of brainstem neurons during both rapid eye movement (REM) and non-REM (NREM) sleep. Crucially, P-waves functionally interact with CA1 activity in a state-dependent manner: during NREM sleep, hippocampal sharp wave-ripples (SWRs) precede P-waves. On the other hand, P-waves during REM sleep are phase-locked with ongoing theta oscillations and are followed by burst firing of CA1 neurons. This state-dependent global coordination between the brainstem and hippocampus implicates distinct functional roles of sleep.

*For correspondence:
shuzo.sakata@strath.ac.uk

Competing interests: The authors declare that no competing interests exist.

## Introduction

The sleep-wake cycle is a fundamental homeostatic process across animal species (*Siegel, 2005*; *Anafi et al., 2019*; *Aulsebrook et al., 2016*). In addition to the physiological functions of sleep (*Boyce et al., 2017*; *Brown et al., 2012*; *Sara, 2017*; *Siegel, 2005*; *Stickgold et al., 2001*; *Imeri and Opp, 2009*; *Liu and Dan, 2019*; *Rasch and Born, 2013*; *Tononi and Cirelli, 2014*), the abnormalities in the sleep-wake cycle are associated with various diseases and disorders (*Mander et al., 2017*; *Musiek and Holtzman, 2016*; *Irwin, 2015*; *Brown et al., 2012*).

Sleep states are typically classified into two major states, non-rapid eye movement (NREM) sleep and REM sleep. While numerous brain regions and cell-types have been identified as part of sleep-regulating circuits (*Brown et al., 2012*; *Héricé et al., 2019*; *Jouvet, 1962*; *Moruzzi, 1963*; *Peever and Fuller, 2017*; *Scammell et al., 2017*; *Weber and Dan, 2016*; *Luppi et al., 2017*; *Adamantidis et al., 2007*; *Tsunematsu et al., 2014*; *Weber et al., 2015*; *Zhang et al., 2019*), sleep-related neural firing and oscillations have also been described across cortical and subcortical regions (*McCarley and Hobson, 1971*; *Hobson et al., 1975*; *Sakai, 1985*; *Steriade, 2006*; *Brown et al., 2012*; *Rasch and Born, 2013*; *Buzsáki, 2015*; *Weber et al., 2015*; *Scammell et al., 2017*; *Weber et al., 2018*; *Héricé et al., 2019*; *Liu and Dan, 2019*). For example, cortical slow oscillations, sleep spindles and hippocampal sharp wave-ripples (SWRs) are prominent neural events during NREM sleep, whereas theta oscillations and ponto-geniculo-occipital (PGO) or pontine (P) waves

**eLife digest** Though almost all animals sleep, its exact purpose remains an enigma. This is particularly true for the period of sleep where people dream most vividly, which is known as rapid eye movement sleep or REM sleep for short. In addition to the eye movements that give it its name, during this phase of sleep, the pupils of the eyes become smaller, muscles relax and neurons in part of the brain activate in a regular, repeating way known as pontine waves or P-waves.

The brainstem is a key brain region that helps the body determine when it is time to sleep and when it is time to be awake. It is found at the back of the brain, and connects the brain to the spinal cord, serving as a conduit for nerve signals to and from the rest of the body. However, it was not clear how the brainstem's activity during sleep interacts with other brain regions that are important in the sleep process, such as the hippocampus.

REM sleep is not unique to humans; in fact, it occurs in all mammals. Tsunematsu et al. studied mice to better understand the role of the brainstem during sleep. In the experiments, the brain waves, muscle tone and pupil sizes of the mice were monitored, while a probe inserted into the brainstem of the mice measured the activity of the neurons. Analysis of the probe data could predict changes in pupil size ten seconds beforehand and transitions between wakefulness, REM sleep and non-REM sleep up to sixty seconds in advance. This long timescale suggests that there are a number of complex interactions following brainstem activity that lead to the changes in sleep state.

Tsunematsu et al. were also able to detect P-waves for the first time in mice and found that they are timed with activity from the hippocampus depending on the sleep state. During REM sleep, the P-waves precede the hippocampal activity, while during non-REM sleep, they follow it. These results further imply that the two sleep states serve different purposes. The detection of P-waves in mice shows that they are similar to other mammals that have previously been studied. Further studies in mice could help to provide more insight into the mechanisms of sleep and the purpose of the different stages.

are seen during REM sleep (*Steriade, 2006*; *Montgomery et al., 2008*; *Buzsáki, 2015*; *Buzsáki, 2002*; *Jouvet, 1969*; *Steriade et al., 1993b*; *Callaway et al., 1987*; *Datta, 1997*; *Rasch and Born, 2013*; *Bizzi and Brooks, 1963*). Although neural ensemble dynamics underlying these sleep-related neural events in the thalamus and the cortex including the hippocampus have been well described (*Steriade, 2006*; *Buzsáki, 2015*; *Buzsáki, 2002*; *Steriade et al., 1993a*), little is known about population activity within the brainstem. To achieve a better understanding of the functional roles of sleep states, it is essential to characterize state-dependent changes in brainstem network activity and their functional interactions with cortical regions across sleep states.

The brainstem, including the midbrain, pons and medulla has long been implicated in the sleep-wake cycle (*Jouvet, 1962*; *Saper et al., 2010*; *Brown et al., 2012*; *Rasch and Born, 2013*; *Weber et al., 2015*; *Weber and Dan, 2016*; *Luppi et al., 2017*; *Scammell et al., 2017*; *Héricé et al., 2019*; *Liu and Dan, 2019*). It contains various nuclei, each of which consists of diverse cell-types and exhibits state-dependent firing (*Brown et al., 2012*; *Rasch and Born, 2013*; *Weber et al., 2015*; *Weber and Dan, 2016*; *Luppi et al., 2017*; *Scammell et al., 2017*; *Héricé et al., 2019*; *Liu and Dan, 2019*; *Weber et al., 2018*; *Zhang et al., 2019*). However, how brainstem populations act in concert is a question that remains poorly explored. For example, the extent to which their activity exhibits anticipatory dynamics for ongoing vigilant states is still unclear. In addition, it is also unclear whether and how brainstem populations functionally interact with various neural oscillations or events in the cortex across sleep states. Characterizing these physiological properties is crucial to uncover the roles of brainstem populations in sleep regulation and ultimately in the functions of sleep states.

In the present study, we adopt several in vivo electrophysiological approaches in mice to investigate state-dependent ensemble dynamics in the brainstem, mainly the pons. We show that, on a timescale of seconds to minutes, brainstem neurons show state-dependent firing with cell type-specificity. They also have a longer predictive power for vigilance states than hippocampal CA1 neurons. On a timescale of sub-seconds, we find state-dependent functional interactions between the brainstem and the hippocampus, with a focus on P-waves. During NREM sleep, the timing of P-waves is

phase-locked with various cortical oscillations and hippocampal SWRs precede P-waves. During REM sleep, P-waves co-occur with hippocampal theta oscillations and precede burst firing of hippocampal neurons. These results imply that brainstem populations not only play a regulatory role in the sleep-wake cycle, but also contribute to global state-dependent dynamics across brain regions.

## Results

### Brainstem population recording across sleep-wake cycles

To investigate the state-dependency of brainstem population activity, we inserted a silicon probe into the mouse brainstem in a head-fixed condition, and performed simultaneous monitoring of cortical electroencephalograms (EEGs), electromyograms (EMGs) and pupil dilation (*Figure 1*). Recorded regions spanned across multiple nuclei, including the sublaterodorsal nucleus, pontine reticular nucleus, medial preoptic nucleus, parabrachial nucleus, pontine central gray, laterodorsal tegmental nucleus and other surrounding areas according to post-mortem histological analysis (*Figure 1—figure supplement 1*). Although a majority of neurons were recorded from the pons, we refer to recorded populations as 'brainstem' neurons because some cells were located in the midbrain and medulla, but not the hypothalamus.

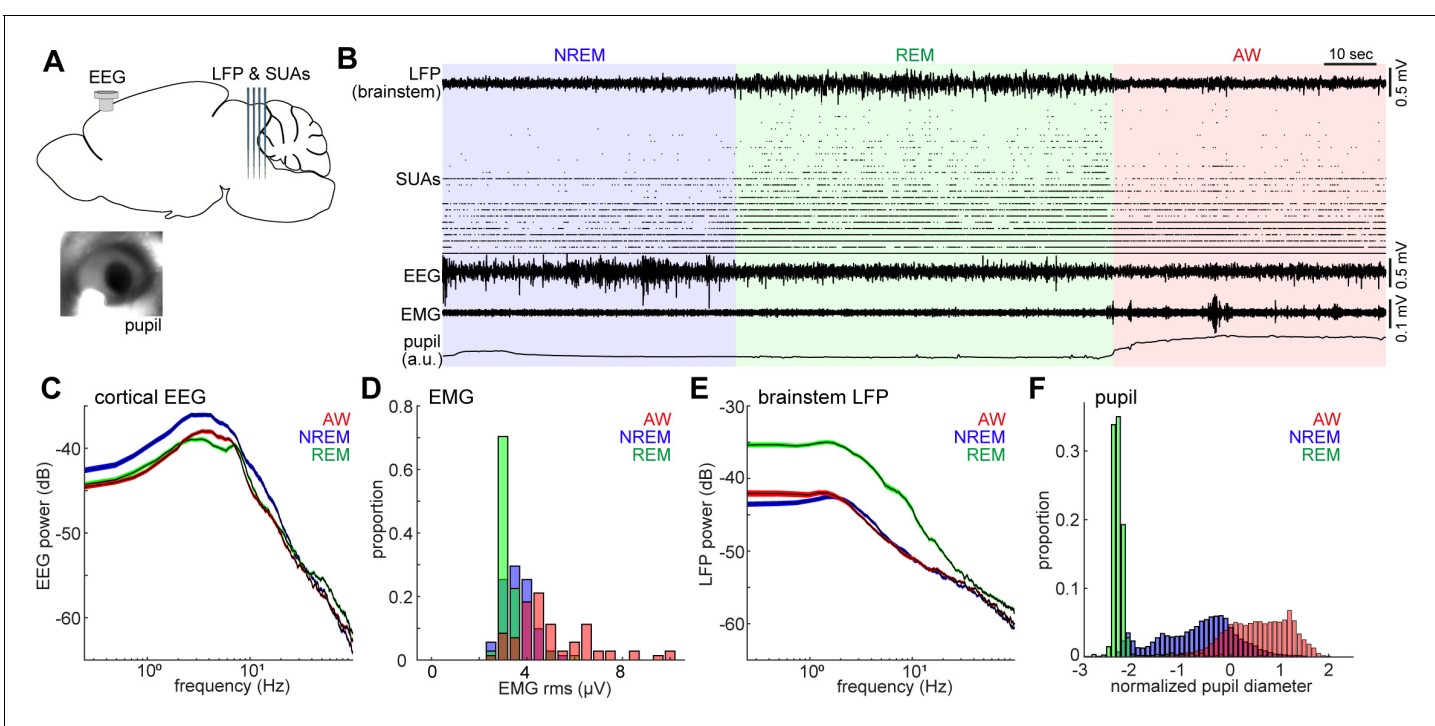

**Figure 1.** Population activity in the brainstem across the sleep-wake cycle. (A) A diagram of the experimental approach, showing the insertion of a silicon probe for extracellular recording in the brainstem and a screw for cortical EEG recording. Pupil dilation and EMGs were also monitored in a head-fixed condition. (B) An example of multiple electrophysiological readings across three behavioral states, including local field potentials (LFPs) in the brainstem (locally subtracted LFP signals), brainstem single unit activities (SUAs), cortical electroencephalograms (EEGs), electromyograms (EMGs) and normalized pupil diameter. REM, rapid eye movement sleep; NREM, non-REM sleep; AW, wakefulness. (C, E) Power spectrum density of cortical EEGs (C) and brainstem LFPs (E) across three behavioral states. Spectra were computed during every 4 s window. Errors indicate SEM. (D, F) Distribution of EMG signals (root mean square) (D) and normalized pupil diameter (F) across three behavioral states. Pupil diameter was normalized as z-score.

The online version of this article includes the following figure supplement(s) for figure 1:

**Figure supplement 1.** Histological analysis.

**Figure supplement 2.** Power spectrum density of cortical EEGs (A) and brainstem LFPs (B) across all recording sessions.

**Figure supplement 3.** Comparisons of sleep architecture, cortical EEGs and EMGs between head-fixed (n = 24) and unfixed conditions (n = 12).

The sleep-wake cycle was classified on the basis of cortical EEGs and EMGs in every 4-s window. On the basis of the classified states, we observed clear state-dependency across measurements (*Figure 1C–F*): wakefulness was characterized by high muscle tone (*Figure 1D*) and pupil dilation (*Figure 1F*), whereas NREM sleep was characterized by higher power of slow oscillations (*Figure 1C*) and a wider dynamic range of pupil diameter (*Figure 1F*). REM sleep was distinct from the other states, with respect to prominent theta oscillations (*Figure 1C*), low muscle tone (*Figure 1D*), higher brainstem LFPs power (*Figure 1E*) and fully constricted pupil (*Figure 1F*). The higher power of brainstem LFPs during REM sleep was preserved across all animals (seven animals, nine recordings) (*Figure 1—figure supplement 2*). Although most of our recordings were performed under a head-fixed condition, the sleep architecture, cortical EEGs and EMGs were generally comparable with those in a tethered unfixed condition (*Figure 1—figure supplement 3*).

Neuronal spiking activity in the brainstem also demonstrated rich state-dependent properties (*Figure 1B*). For example, a subset of neurons fired exclusively during REM sleep, indicating state-dependent population firing on a timescale of second-to-minute. In addition, we also observed frequent burst firing across neurons on a sub-second timescale during REM sleep. In the following analysis, we investigate state-dependent brainstem neural ensembles on two distinct timescales: a long timescale of seconds to minutes (*Figures 2–4*) and a sub-second timescale (*Figures 5–7*).

## Diversity and specificity of state-dependent neural activity in the brainstem

To assess the state-dependent firing of individual neurons in the brainstem on a timescale of seconds to minutes, we performed in vivo silicon probe recording (*Figure 2A*) from seven head-fixed mice (nine recording sessions) and examined how individual neurons change their firing across behavioral states. *Figure 2B* shows representative examples of state-dependent firing from four simultaneously recorded neurons. Even within a particular state in the same animal, brainstem neurons show highly diverse and dynamic firing.

To classify neurons according to their state-dependent firing, we computed mean firing rate in each state across neurons (n = 76) and applied a hierarchical clustering algorithm (*Figure 2C*). We identified four functional classes: awake (AW)-on neurons (23.7%) were more active during wakefulness than during sleep states; REM-off neurons (17.1%) reduced their firing during REM sleep; REM/AW-on neurons (6.6%) were quiet during NREM sleep; and the largest class (52.6%) was REM-on neurons, which showed the highest firing rate during REM sleep. Thus, we confirmed highly diverse state-dependent firing in the brainstem.

Because the recorded neurons were distributed across various nuclei in the brainstem, it was difficult to determine their state-dependency in each nucleus. However, a subset of neurons was probably recorded from the cholinergic system, namely the pedunculopontine tegmental nucleus and the laterodorsal tegmental nucleus, which showed AW-on or REM-on activity (*Figure 2—figure supplement 1*). To verify this, we performed in vivo fiber photometry of Ca$^{2+}$ signals from pontine cholinergic neurons by expressing GCaMP6s in freely behaving mice (four animals, 12 recording sessions) (*Figure 2D*). Consistent with the data from in vivo electrophysiology, cholinergic populations showed larger activity during REM sleep and wakefulness than during NREM sleep ($F_{2,32}$ = 5.12, p=0.012, one-way ANOVA) (*Figure 2E–G*). Therefore, although the state-dependency of individual neuronal firing in the brainstem is diverse, we also confirmed state-dependent and cell-type-specific firing.

## Behavioral correlates of the sleep-wake cycle and underlying neural activity in the brainstem

Pupil diameter is an excellent biomarker of global brain state or arousal level (*Aston-Jones and Cohen, 2005*; *Yüzgeç et al., 2018*; *Larsen and Waters, 2018*; *McGinley et al., 2015*) and activity in brainstem neurons, especially in locus coeruleus norepinephrine neurons, correlates with pupil diameter (*Aston-Jones and Cohen, 2005*). However, it is still unclear how pupil dilation changes around the transition of sleep-wake states and to what extent the activities of brainstem neurons as a population can predict pupil dilation quantitatively. To address these issues, we analyzed datasets from head-fixed mice either with silicon probe recordings from the brainstem (six animals, six recording sessions) or the hippocampus (two animals, three recording sessions), or with field potential recording from the brainstem with a bipolar electrode (six animals, nine recording sessions).

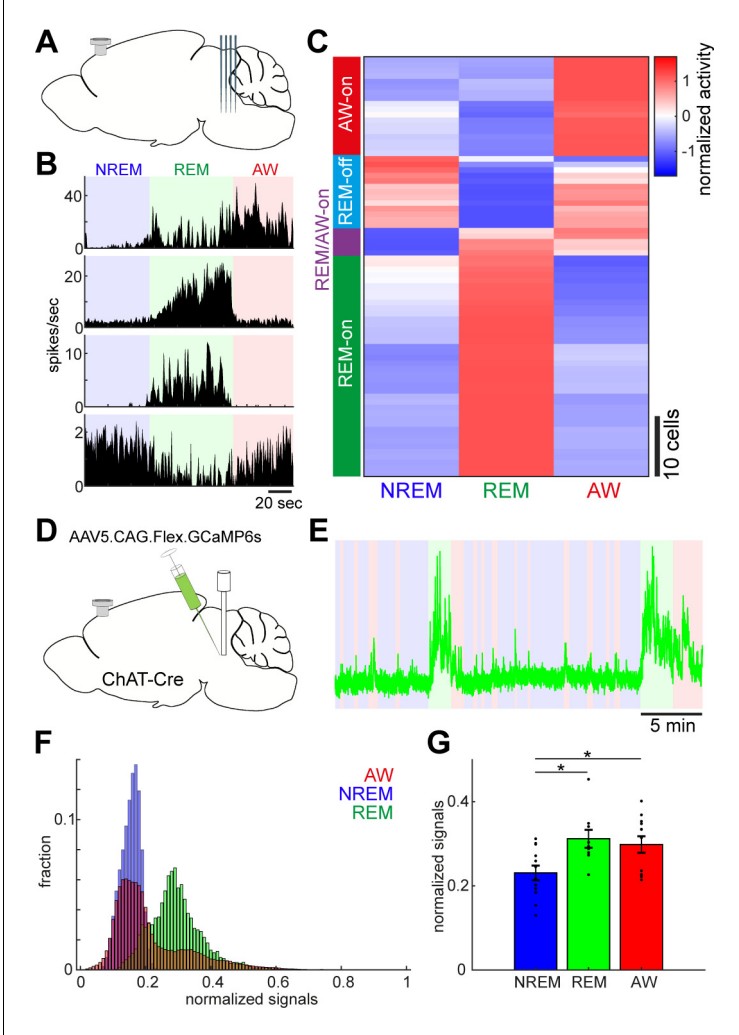

**Figure 2.** Diverse and cell-type-specific state-dependent firing in brainstem neurons. (A) A diagram of the experimental approach, showing a silicon probe and a cortical EEG electrode. (B) Four examples of simultaneously recorded neurons. (C) Classification of functional classes. Firing rates across the three behavioral states were normalized as z-score for individual cells and hierarchical clustering was then applied. (D) A diagram of the experimental approach for fiber photometry-based $Ca^{2+}$ imaging from pontine cholinergic neural populations in a freely behaving condition. (E) An example of fluorescent signals across sleep-wake cycles. Fluorescent signals (470 nm) were normalized by off-peak (405 nm) signals. Red, wakefulness; blue, NREM sleep; green, REM sleep. (F) Distributions of fluorescent signals across the three behavioral states. (G) Group statistics of average fluorescent signals from 12 recordings from four mice ($F_{2,32}$ = 5.12, p=0.012, one-way ANOVA). *, p<0.05 with post-hoc Tukey's honest significant difference criterion.

The online version of this article includes the following figure supplement(s) for figure 2:

**Figure supplement 1.** Neurons recorded from pontine cholinergic nuclei and their state dependency.

As previously reported (*Yüzgeç et al., 2018*), head-fixed mice kept their eyes open, allowing us to monitor pupil dilation across states along with cortical EEG and EMG. The effects of behavioral states on pupil diameter was statistically significant (*Figure 3A*, $F_{2, 53}$ = 220.33, p<0.0001, one-way ANOVA), with pupil diameter being constricted during REM sleep and dilated during wakefulness.

With respect to pupil dilation dynamics across states (*Figure 3B* and *Figure 3—figure supplement 1*), pupil diameter dynamically fluctuated during wakefulness and gradually constricted during NREM sleep. Typically, 10–20 s before REM sleep, the pupil diameter would further decrease and was fully constricted during REM sleep (*Figure 3—figure supplement 1*).

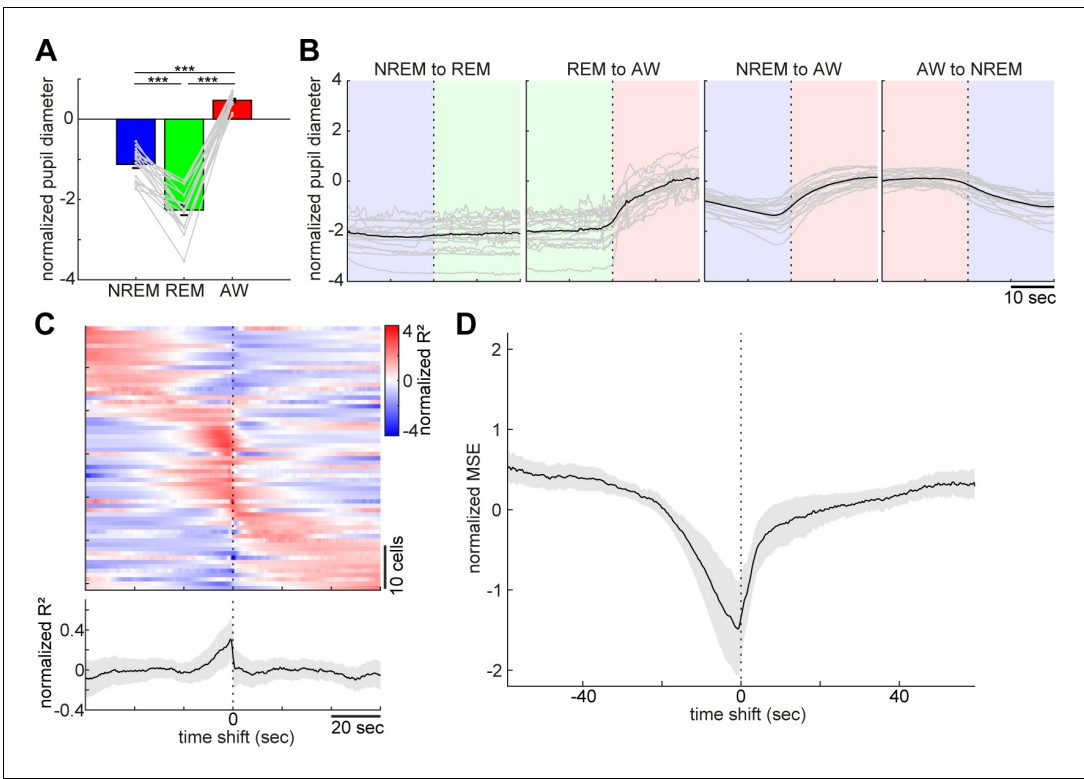

**Figure 3.** Pupil dilation across the sleep-wake cycle and prediction of pupil dilation by activity of brainstem populations. (A) Mean normalized (z-scored) pupil diameter across the sleep-wake cycle (n = 18, $F_{2,\ 53}$ = 220.33, p<0.0001. one-way ANOVA). ***, p<0.0001, with post-hoc Tukey's honest significant difference criterion. (B) Pupil dilation during sleep-wake state transitions (n = 18). (C) Linear regression analysis to predict pupil diameter with individual neuronal activity. (*Top*) Normalized (z-scored) cross-validated $R^2$ values are color-coded across neurons. (*Bottom*) The average of normalized cross-validated $R^2$ values. Error, SEM. (D) Multiple linear regression analysis to predict pupil diameter according to simultaneously recorded neuronal activity. The average of normalized (z-scored) root mean square errors across datasets (n = 6). Error, SEM.

The online version of this article includes the following figure supplement(s) for figure 3:

**Figure supplement 1.** Pupil dilation and eye movement across vigilance states.

Taking advantage of the simultaneous recording of neural population activity and pupil monitoring, we examined how brainstem neurons can predict pupil dilation. First, we predicted pupil dilation on the basis of the activity of individual neurons (*Figure 3C*) by applying a linear regression analysis. Because it was expected that the preceding neural activity can better predict pupil dilation, we systematically shifted the temporal relationship between spike trains and pupil diameter (see 'Materials and methods'). As expected, most of the neurons showed asymmetric profiles of $R^2$ values (*Figure 3C*). Although individual profiles were diverse, the average profile showed brainstem neuron activities that were predictive of pupil diameter around 10 s in advance. We also predicted pupil diameter on the basis of simultaneously recorded brainstem neurons (*Figure 3D*) by applying a multiple linear regression analysis. As with individual neurons, we observed an asymmetric profile of predictability. Thus, changes in brainstem neural activity preceded those in pupil diameter. Thereby, brainstem populations have predictive power for pupil diameter.

## Longer predictability of brainstem ensembles for vigilance states

Next, we examined whether and to what extent brainstem neurons can predict sleep-wake states. To address this, first, we extracted the features of neural population activity by applying non-negative matrix factorization (NMF), a technique that finds compact and easily interpretable representations of neural population activity in the form of parts-based activity patterns (*Lee and Seung, 1999*;

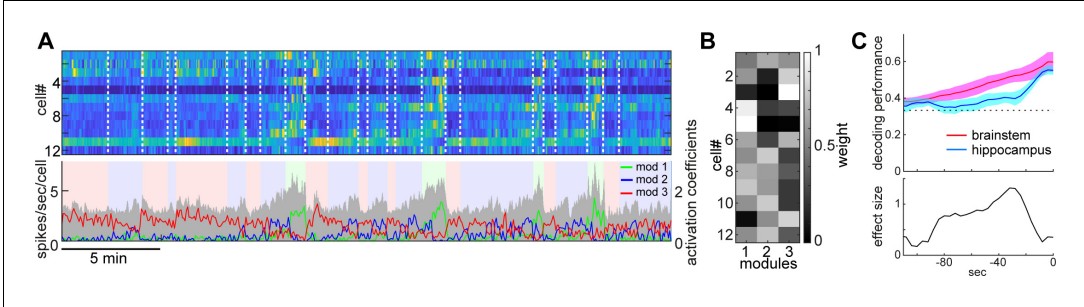

**Figure 4.** State-dependent brainstem population dynamics and their ability to predict vigilance states. (**A**) Simultaneously recorded brainstem neurons across states and modules extracted with non-negative matrix factorization (NMF). (*Top*) Firing profiles of brainstem neurons. Firing rate was normalized by the maximum firing rate and the normalized values were color-coded. Dotted lines indicate state transitions. (*Bottom*) Average population activity (gray) and activation coefficients for each module derived by NMF. Background colors indicate behavioral states (red, AW; blue, NREM; green, REM). (**B**) Weights across neurons for each module. (**C**) Decoding of behavioral states from population activity. (*Top*) Decoding performance of brainstem and hippocampal CA1 neural populations for behavioral states as a function of time-shift. Error, SEM. (*Bottom*) Effect size as a function of time-shift.

The online version of this article includes the following figure supplement(s) for figure 4:

**Figure supplement 1.** Weights of each module derived from non-negative matrix factorization (**A**) and individual firing profiles (**B**).

**Figure supplement 2.** Histological analysis of dorsal hippocampal recording.

---

*Onken et al., 2016*) (*Figure 4A and B*) (see 'Materials and methods'). Overall firing rate reflected state changes (*Figure 4A*), whereas NMF was able to extract several modules that captured state-dependent firing patterns across neurons in an unsupervised fashion. For example, module 1 represented REM-on activity, whereas module 2 was activated at the end of NREM sleep and module 3 was most active during wakefulness. Indeed, the weights in each module were consistent with the state-dependency of individual neural firing (*Figure 4—figure supplement 1*).

Besides modules capturing firing patterns across neurons, NMF also yielded the activation coefficients of these modules. We noticed that the dynamics of these activation coefficients show predictive activity: in the case of *Figure 4A*, the activation coefficients of modules 1 and 2 gradually built up during REM and NREM sleep, respectively. Therefore, we hypothesized that brainstem population activity exhibits not just state-dependency, but also predictive power for behavioral states (i.e., wakefulness, NREM sleep and REM sleep). To test this, we took the activation coefficient profiles from the three modules and classified behavioral states by training a linear classifier, with systematic time shifting (*Figure 4C*). Brainstem populations showed predictive activity tens of seconds before transitions into all three behavioral states.

Does this predictive activity last longer in brainstem populations than in other brain regions? Of many brain regions, hippocampal neurons are well known to exhibit state-dependent firing (*Buzsáki, 2002*; *Buzsáki, 2015*; *Mizuseki and Miyawaki, 2017*), but they are less likely to contribute to the regulation of the sleep-wake cycle directly. Hence, hippocampal neurons could serve as a reference to assess the ability of brainstem populations to predict behavioral states. To examine whether brainstem populations exhibit predictive activity over a longer period than do hippocampal populations, we performed the same analysis for neural activity in the hippocampal CA1 (*Figure 4C*). Although CA1 neurons also had predictive power for several tens of seconds, the profile was relatively short-lasting compared to that of brainstem neurons. Thus, brainstem neurons have longer-lasting predictive power for behavioral states than hippocampal CA1 neurons.

## Brainstem population activity underlying P-waves during NREM and REM sleep

On a timescale of seconds to minutes, brainstem neurons show diverse but specific state-dependent firing and have predictive power for pupil dilation and behavioral states. To investigate brainstem neural firing on a sub-second timescale, we focused on P-waves (*Callaway et al., 1987*;

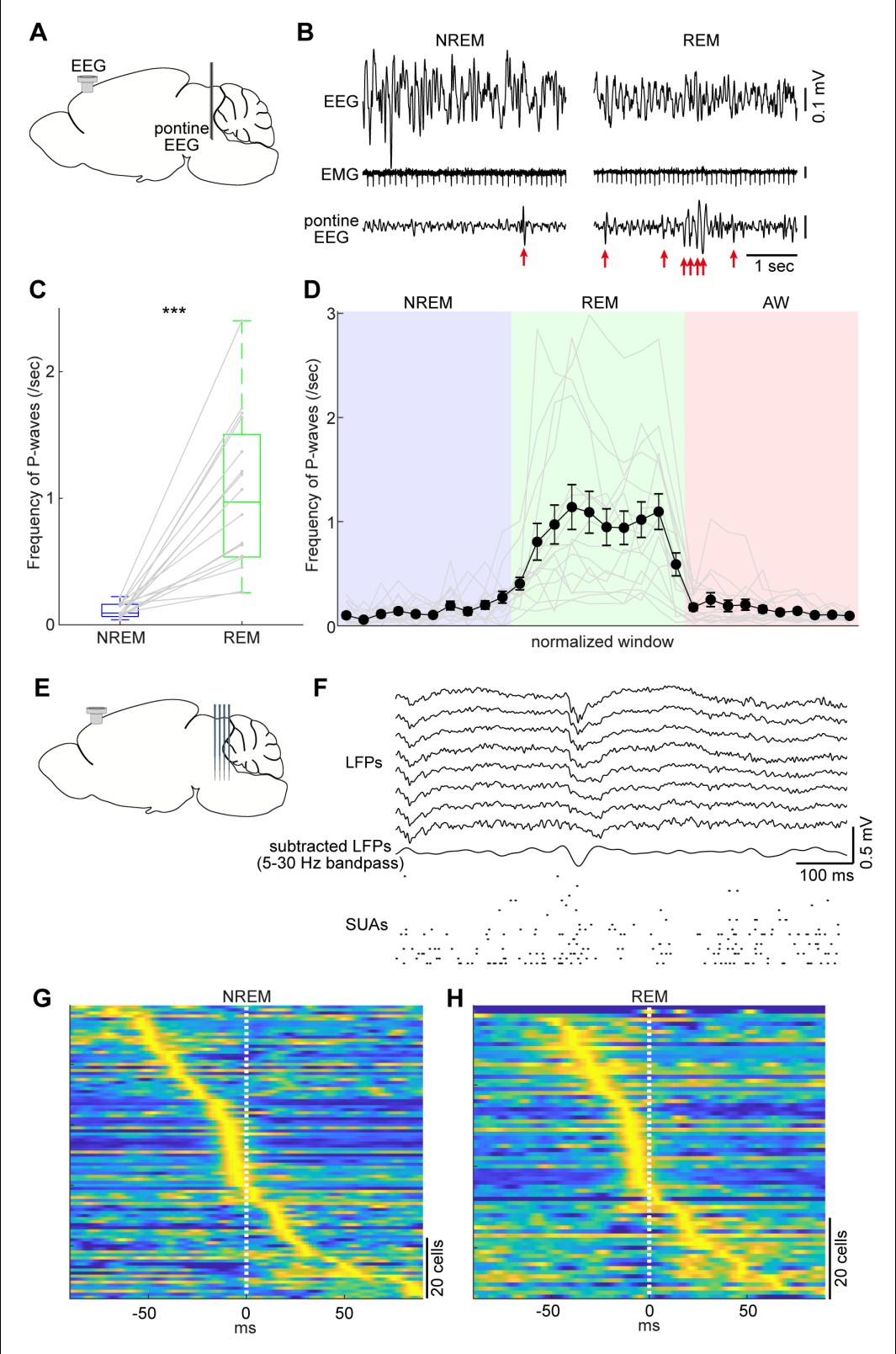

**Figure 5.** Pontine waves (P-waves) in the mouse. (**A**) A diagram of the experimental approach, showing a bipolar electrode in the pons and a cortical EEG electrode. (**B**) Examples of P-waves during NREM (left) and REM sleep (right), showing cortical EEG and EMG traces. (**C**) Frequency of P-waves during NREM and REM sleep (n = 16). ***, p<0.0001, two-tailed *t*-test. (**D**) Temporal evolution of P-wave frequency. The duration of each state episode was
*Figure 5 continued on next page*

*Figure 5 continued*

normalized to one. Error, SEM. (**E**) A diagram of the experimental approach, showing a multi-shank silicon probe in the brainstem and a cortical EEG electrode. (**F**) An example of a P-wave, showing subtracted LFPs from a probe shank, and multiple single unit activity. (**G, H**) Pooled peri-event time histograms of brainstem single units relative to P-wave timing during NREM (**G**) and REM sleep (**H**). Time zero is the timing of P-waves (trough time). Each peri-event time histogram is color-coded by normalizing the maximum firing rate for each cell. The order of single units was sorted by the peak timing in each state.

*Datta, 1997*). Although these sub-second neural events in the brainstem have long been recognized, the underlying neural ensembles and relations to other sleep-related oscillations are not fully understood.

Taking advantage of our dataset, we first examined whether the mouse brainstem exhibits P-waves like those observed in other mammalian species. We implanted a bipolar electrode in the pons (n = 16 recordings) (*Figure 5A*) and monitored LFPs by subtracting signals. During REM sleep, we observed large-amplitude irregular neural events, which often appeared bursts (*Figure 5B* right). We also observed similar, but isolated, neural events during NREM sleep (*Figure 5B* left). These neural events appeared more often during REM sleep (p<0.0001, two-tailed *t*-test) (*Figure 5C*). Intriguingly, the frequency of these events gradually increased during NREM to REM sleep transitions and decreased during REM sleep to wakefulness transitions (*Figure 5D*). Because these characteristics generally resemble those in other species (*Callaway et al., 1987*; *Datta, 1997*), we concluded that these neural events are P-waves in mice.

P-waves can also be seen in silicon probe recordings (*Figure 5E*). Similar large-amplitude, irregular neural events were observed in subtracted and filtered LFPs (*Figure 5F*). Many of the simultaneously recorded brainstem neurons fired during P-waves. To assess this tendency, we pooled the peri-event firing profiles of all recorded brainstem neurons around P-waves (*Figure 5G,H*). The firing profiles were aligned at the trough timing of P-waves. A subset of neurons showed peak firing at the falling phase of P-waves. This tendency was consistent between NREM and REM sleep with respect

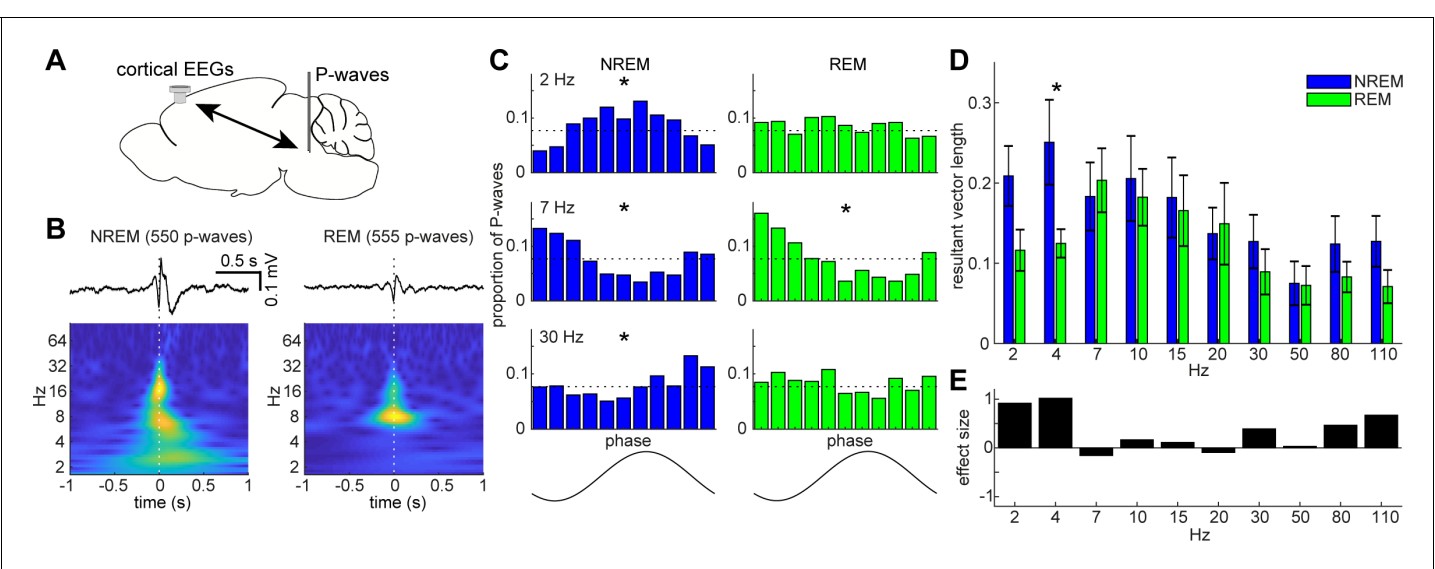

**Figure 6.** State-dependent interactions between P-waves and cortical oscillations. (**A**) A diagram of an experimental approach showing a bipolar electrode in the pons and a cortical EEG electrode. (**B**) Examples of averaged event-triggered cortical EEGs and scalograms during NREM (left) and REM sleep (right). Time zero is the timing of P-waves (trough time). (**C**) Examples of phase-histograms. Cortical EEGs were filtered at certain frequency bands and the proportion of P-waves elicited in each phase bin was calculated. *, p<0. 01, Rayleigh test. (**D**) Phase modulations across frequency bands of cortical EEGs. Resultant vector length represents the mean measurement of circular spread of the phase relationship between cortical oscillations and P-waves. The effect of sleep states on resultant vector length was significant ($F_{1,179}$ = 4.90, p<0.05, two-way ANOVA). *, p<0.05 (post-hoc Tukey's hones significant difference test). Error, SEM. (**E**) Effect size of states across frequency bands.

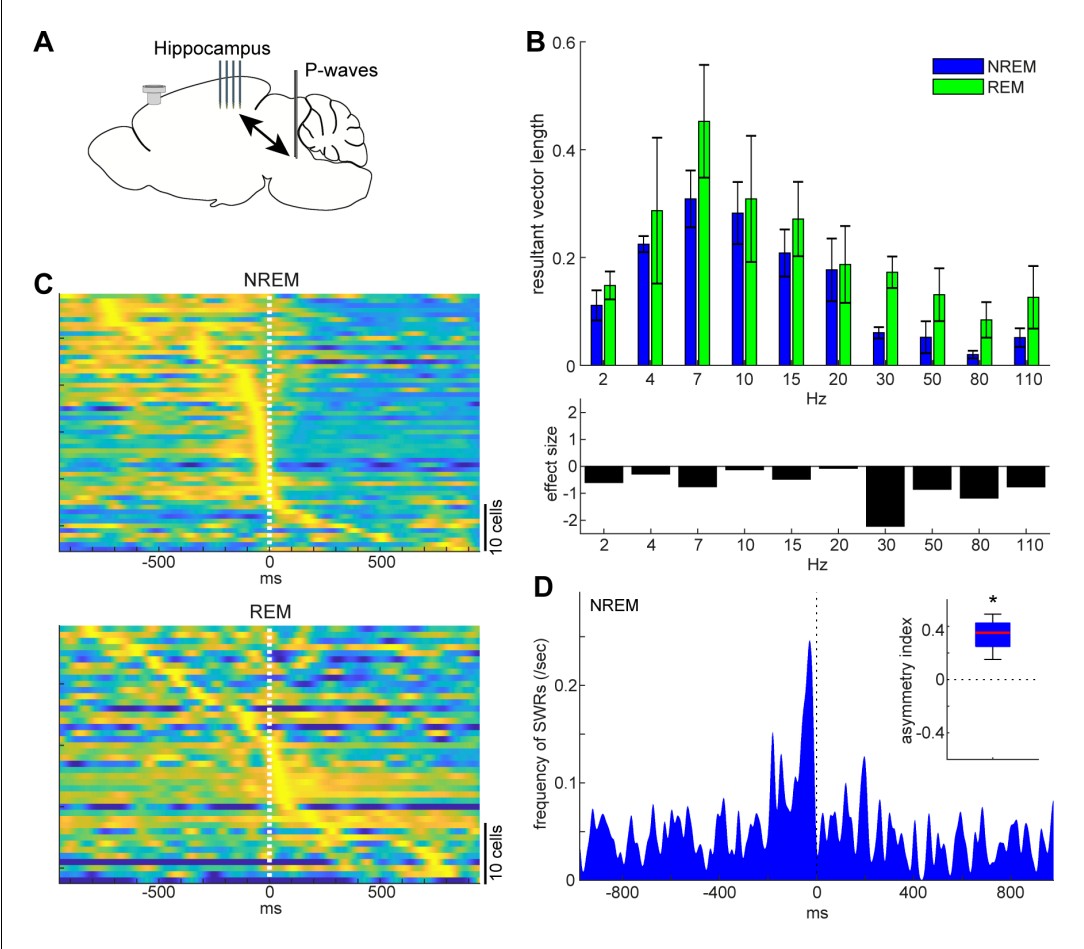

**Figure 7.** State-dependent interactions between P-waves and hippocampal activity. (**A**) A diagram of the experimental approach showing a multi-shank silicon probe in the hippocampus CA1, a bipolar electrode in the pons and a cortical EEG electrode. (**B**) Phase modulations across frequency bands of hippocampal LFPs (top) and effect size of states (bottom). The effect of sleep states on resultant vector length was significant ($F_{1,79} = 4.00$, $p<0.05$, two-way ANOVA). Error, SEM. (**C**) Pooled peri-event time histograms of CA1 single units relative to P-wave timing in NREM (top) and REM sleep (bottom). Time zero is the timing of P-waves (trough time). Each peri-event time histogram is color-coded by normalizing the maximum firing rate for each cell. The order of single units was sorted by the peak timing in each state. (**D**) Frequency of hippocampal sharp wave-ripples (SWRs) relative to P-wave timing in NREM sleep. Time zero is the timing of P-waves (trough time). After binning at 1 ms resolution, the histogram was smoothed with a 50 ms Gaussian kernel. *Inset*, asymmetry index provides the normalized difference in the number of SWRs before and after P-waves in a 100 ms window. Larger positive values mean that more SWRs occurred before the P-waves. *, $p<0.05$ (t-test, n = 4).

to neural firing within the brainstem, suggesting that P-waves during NREM sleep (P-waves[NREM]) are equivalent to P-waves during REM sleep (P-waves[REM]).

## State-dependent functional interactions between P-waves and cortical activity

The co-firing of a subset of brainstem neurons underlies P-waves during both NREM and REM sleep. What are the impacts of such neural events on other brain regions? Are P-waves associated with any other sleep-related neural events? Addressing these questions would provide insight into the functions of P-waves. To this end, first, we investigated the relationship between P-waves and cortical EEGs (*Figure 6*). During NREM sleep, P-waves were associated with multiple oscillatory components. Averaged P-wave-triggered cortical EEGs exhibited multiple phasic components (*Figure 6B*), which consisted of delta (1–4 Hz), theta (~7 Hz) and beta (15 ~ 30 Hz) frequencies. To examine this trend further, we assessed the phase relationship between P-wave timing and cortical oscillations (*Figure 6C*). We found significant phase preferences of P-wave timing (p<0.01, Rayleigh test). We further assessed this phase-locking activity by computing resultant vector length (*Figure 6D*), which

is a descriptive circular statistic and represents the length of the mean resultant vector (*Berens, 2009*): the closer this value is to one, the more concentrated is the phase coupling. We found larger phase modulation at delta and beta ranges.

On the other hand, P-waves[REM] exhibited distinct associations with cortical oscillations (*Figure 6B–D*). We observed significant phase modulation at theta range (p<0.01) (*Figure 6C*), indicating that two prominent neural markers in REM sleep, that is, theta oscillations and P-waves, are temporally organized. We found that the effect of sleep states on these phase-locking activities was statistically significant ($F_{1,179} = 4.90$, p<0.05, two-way ANOVA). We also computed effect size (*Figure 6E*). The effect was larger at the delta range during NREM sleep. Hence, the temporal coupling between P-waves and cortical oscillations was modified by sleep states.

Next, we investigated the relationship between P-waves and hippocampal CA1 activity (*Figure 7*). We started by assessing the phase relationship between hippocampal LFPs and P-waves across frequency bands (*Figure 7B*). We found that the effect of sleep states on the resultant vector length was significant ($F_{1,79} = 4.00$, p<0.05, two-way ANOVA). The timing of P-waves was strongly phase-locked at the theta range (~7 Hz) in both sleep states and there was no significant difference in the resultant vector length across frequencies, but we observed stronger phase modulations in high frequency ($\geq$30 Hz) components during REM sleep based on the effect size. We further examined underlying spiking activity in the hippocampus CA1 (*Figure 7C*). Intriguingly, although a subset of CA1 neurons fired most strongly around the timing of P-waves during both NREM and REM sleep, the temporal order between CA1 neural firing and P-waves was state-dependent (*Figure 7C*): during NREM sleep, co-firing of hippocampal neurons was followed by P-waves, whereas P-waves were followed by burst firing in a subset of CA1 neurons during REM sleep. To test the hypothesis that co-firing of CA1 neurons during NREM sleep may reflect SWRs, we detected high-frequency ripple events based on hippocampal LFPs to assess the temporal relationship between ripples and P-waves[NREM] (*Figure 7D*). We found that ripple events preceded P-waves during NREM sleep across all of four recordings (*Figure 7D inset*) (p<0.05, *t*-test). Thus, P-waves are strongly associated with hippocampal activity in both sleep states. However, these associations are state-dependent.

## Discussion

Although state-dependent neural ensembles have been intensively characterized in the cortex including the hippocampus, little is known about state-dependent brainstem ensembles. Here, we investigated neural population activity in the brainstem, primarily the pons, on two distinct timescales. On a timescale of seconds to minutes, brainstem neurons show diverse state-dependent firing, with cell-type-specificity in pontine cholinergic neurons. Brainstem activity can collectively predict pupil dilation as well as behavioral states. The ability to predict behavioral states is longer lasting than that of hippocampal neurons. These relatively slow dynamics may be related to observations from optogenetic experiments in which the effect of optogenetic stimulation on state transitions often emerges tens of seconds after stimulus onset (*Adamantidis et al., 2007*; *Van Dort et al., 2015*; *Zhang et al., 2019*; *Tsunematsu et al., 2013*; *Tsunematsu et al., 2014*).

On a timescale of sub-seconds, we characterized P-waves in the mouse, with respect to underlying neural firing as well as associated neocortical and hippocampal activities. P-waves typically appear during REM sleep and less often during NREM sleep. P-waves in both sleep states are accompanied by synchronous firing of brainstem neurons, suggesting that underlying local activity during P-waves is similar between sleep states. However, the relationship of P-waves to cortical neural events is state-dependent: the timing of P-waves[NREM] are phase-locked to various cortical oscillations and hippocampal SWRs precede P-waves[NREM], suggesting that P-waves[NREM] are part of the brain-wide neural events triggered by SWRs. During REM sleep, P-waves are phase-locked most strongly at theta frequency in both the neocortex and hippocampus. Crucially, P-waves precede firing in a subset of hippocampal CA1 neurons, suggesting that P-waves[REM] may trigger brain-wide neural events. Thus, P-waves are part of the state-dependent coordinated activity across the brain.

### Technical considerations

State-dependent activity in the brainstem has been described over the past several decades by using various methods (*Pace-Schott and Hobson, 2002*; *Datta and Maclean, 2007*; *Steriade and McCarley, 1990*). The present study utilized a silicon probe to monitor activity from multiple

neurons simultaneously with high temporal resolution. This approach allowed us (1) to quantify the state dependency of brainstem neural ensemble dynamics on a timescale of seconds to minutes, (2) to decode vigilance states on the basis of population activity, and (3) to characterize neural population activity underlying P-waves for the first time. However, because silicon probe recording alone has a limitation in identifying cell types, additional approaches, such as $Ca^{2+}$ imaging (*Figure 2*) or electrophysiology with optogenetic tagging (*Weber et al., 2015*; *Yague et al., 2017*; *Zhang et al., 2019*), can complement this approach in order to determine how specific types of neurons contribute to state-dependent neural ensembles in the brainstem.

## Slow dynamics of brainstem ensemble dynamics

Our results in *Figures 3* and *4* are consistent with the notion that brainstem populations play a regulatory role in pupil dilation and constriction (*Aston-Jones and Cohen, 2005*; *Larsen and Waters, 2018*), as well as in global brain states (*Brown et al., 2012*; *Héricé et al., 2019*; *Luppi et al., 2012*; *Steriade et al., 1984*; *Weber and Dan, 2016*). Crucially, the asymmetric profile of the predictability of pupil diameter suggests that the modulation of brainstem activity precedes pupil dilation, rather than being simply correlated to it.

The long-lasting predictability of brainstem populations for behavioral states is not trivial. Intriguingly, the slow (30–60 s) timescale recalls us the timescale observed in some of optogenetic experiments: although optogenetic stimulation can modulate neural firing at a millisecond resolution, the effect of optical stimulation on state transitions typically emerges tens of seconds after stimulus onset (*Adamantidis et al., 2007*; *Van Dort et al., 2015*; *Zhang et al., 2019*). The exact mechanism is still unknown, but we hypothesize that the modulation of neural activity in the brainstem occurs tens of seconds before global brain state transitions from one state to another. In other words, each state emerges from complex interactions across various regions of the brain.

## P-waves in mice

Although PGO or P-waves have been studied since the 1960s in several mammalian species (*Callaway et al., 1987*; *Datta, 1997*; *Bizzi and Brooks, 1963*), to the best of our knowledge, we are the first to characterize P-waves in mice. Given the growing importance of mice as an animal model for sleep research (*Héricé et al., 2019*), the confirmation of P-waves in mice is important for further interrogation with respect to the generation mechanism and function of P-waves.

We have noticed several similarities and differences in P-waves between mice and other species. First, the waveform of P-waves in mice is generally consistent with those in other species, such as cats (*Callaway et al., 1987*; *Jeannerod et al., 1965*) and rats (*Datta, 1997*; *Farber et al., 1980*), suggesting that underlying neural ensembles may be similar across species. Indeed, the phasic firing just before (<25 ms before) the trough of P-waves (*Figure 5*) resembles the observations in cats (*McCarley et al., 1978*; *Steriade et al., 1990*; *Sakai and Jouvet, 1980*; *Nelson et al., 1983*). In the near future, it would be interesting to extend our approach further to explore how activity propagates across brainstem neurons during P-waves by identifying cell-types. Second, the frequency of P-waves during REM sleep is generally consistent across species (*Datta, 1997*). However, we have also noticed that the frequency of detected P-waves varied across our experiments. This may be explained by variation in either the REM sleep quality or the electrode positions. Further analysis of P-waves across brainstem nuclei will provide insights into their relationship with sleep homeostasis and the mechanism of P-wave genesis in mice. Datta and his colleagues performed a series of pioneering studies and reported species differences in the site of P-wave genesis between cats and rats (*Datta et al., 1998*; *Datta and Hobson, 1995*; *Datta et al., 1992*): the caudolateral peribrachial area was identified as the induction site of P-waves (PGO waves) in cats (*Datta and Hobson, 1995*; *Datta et al., 1992*), whereas the subcoeruleus nucleus was identified in rats (*Datta et al., 1998*). It will be important to revisit this issue in mice by adopting modern technologies. Third, the temporal evolution of P-wave frequency is generally similar in mice and cats: the frequency of PGO-waves gradually increases before the transition of NREM to REM sleep in cats (*Steriade et al., 1989*). Although it was weak, a similar tendency was observed in our recordings (*Figure 5D*). Rather, the frequency of P-waves increases during REM sleep. This subtle difference may be explained by anatomical differences between species (*Datta, 2012*). These species differences may also imply differences in the function of P-waves between species. Finally, although P-waves appear more frequently

during REM sleep, it is important to note that similar neural events also occur during NREM sleep. Given state-dependent interactions between P-waves and cortical oscillations (*Figures 6* and *7*), the mechanisms of P-wave genesis are probably distinct.

### State-dependent global coordination of neural ensembles

The temporal correlation between P-waves and hippocampal theta rhythms during REM sleep is consistent with previous studies in cats and rats (*Karashima et al., 2002*; *Karashima et al., 2004*; *Karashima et al., 2005*; *Sakai et al., 1973*). The phase-locked activity with fast gamma (80–110 Hz) in the hippocampus may relate to the recent observation that demonstrated the close association between local hippocampal theta and fast gamma events and brain-wide hyperemic events (*Bergel et al., 2018*). Because a number of hippocampal CA1 neurons fire immediately after P-waves (*Figure 7C*), P-waves may play a role in the regulation of hippocampal ensemble dynamics as well as in brain-wide events during REM sleep.

On the other hand, the picture during NREM sleep seems to be distinct. Because SWRs precede P-waves (*Figure 7D*), and because SWRs are known to be generated within hippocampal circuits (*Buzsáki, 2015*), SWRs play a leading role in brain-wide sub-second neural events, including P-waves. These state-dependent brain-wide neural ensembles imply that different sleep states have distinct functional roles.

## Materials and methods

### Animals

A total of 21 mice were used in this study (*Supplementary file 1*) and were housed individually with a 12 hr:12 hr light/dark cycle (light on hours: 7:00–19:00). Mice had ad libitum access to food and water. All experiments were performed during the light period. Their genotypes consisted of wild-type, ChAT-IRES-Cre (JAX006410), or ChAT-IRES-Cre::Ai32 (JAX012569) on a C57BL/6 background. ChAT-IRES-Cre::Ai32 mice were used to identify pontine cholinergic neurons in post-hoc histological analysis. For brainstem silicon probe recordings, six animals were used (nine recordings). For hippocampal silicon probe recordings, two animals were used (four recordings). For P-wave recordings, ten animals were used, but four were excluded because of electrode mispositioning or lack of histological data. 16 datasets were used for further analysis. For pupil monitoring, 14 animals were used, but one animal was excluded because it closed its eyes during recording. 18 datasets were used for further analysis. For fiber photometry, four animals were used (12 recordings). The detailed information of genotypes, age, sex, body weight and the number of recordings is provided in *Supplementary file 1*.

### Surgical procedures

For all in vivo electrophysiological experiments, mice were anesthetized with isoflurane (5% for induction, 1–2% for maintenance) and placed in a stereotaxic apparatus (SR-5M-HT, Narishige). Body temperature was maintained at 37°C with a feedback temperature controller (40–90–8C, FHC). Lidocaine (2%, 0.1–0.3 mg) was administered subcutaneously at the site of incision. Two bone screws were implanted onto the skull as electrodes for cortical EEGs, and twisted wires were inserted into the neck muscle as electrodes for EMG. Another bone screw was implanted onto the cerebellum as a ground/reference.

For pontine EEG recording, bipolar electrodes were bilaterally implanted in the medial parabrachial nucleus of the pons (5.1 mm posterior, 1.2 mm lateral from bregma, 3.2 mm depth from brain surface). The bipolar electrodes consisted of 75 or 100 μm diameter stainless wires (FE631309, Advent Research Materials and FE205850, Goodfellow, respectively). The tip of two glued wires were separated by 0.5–1.0 mm vertically to differentiate EEG signals. All electrodes were connected to connectors (SS-132-T-2-N, Semtec) and securely attached onto the skull with dental cement. A pair of nuts was also attached onto the skull with dental cement as a head-post. After the surgery, Carprofen (Rimadyl, 5 mg/kg) was administered intraperitoneally.

For brainstem or hippocampal silicon probe recording, in addition to bone screws for cortical EEGs and a ground/reference, a pair of nuts was attached onto the skull with dental cement as a head-post. After the head-post surgery, the animals were left to recover for at least 5 days. During

the habituation period, the animals were placed in a head-fixed apparatus, by securing them by the head-post and placing the animal into an acrylic tube. This procedure was continued for at least 5 days, during which the duration of head-fixing was gradually extended from 10 to 120 min. A day after the habituation period, the animals were anesthetized with isoflurane and a craniotomy to insert the silicon probe into the brainstem and hippocampus was performed. Two craniotomies were performed, one on the left hemisphere (4.0 mm to 5.5 mm posterior, 1.0 to 1.3 mm lateral from bregma) for the brainstem recording and a second also on the left hemisphere (2.0 mm posterior, 1.5 mm lateral from bregma) for the hippocampus recording. To protect and prevent the brain from drying, the surface was covered with biocompatible sealants (Kwik-Sil and Kwik-Cast, WPI). The following day, the animals were placed in the head-fixed apparatus for electrophysiological recording as described below.

For fiber photometry experiments, cortical EEG and EMG electrodes were implanted as described above and connected to a 2-by-3 piece connector (SLD-112-T-12, Semtec). Two additional anchor screws were implanted bilaterally over the parietal bone to provide stability and a small portion of a drinking straw was placed horizontally between the anchor screws and the connector. The viral vector (AAV5-CAG-flex-GCaMP6s-WPRE-SV40, Penn Vector Core; titer $8.3 \times 10^{12}$ GC/ml) was microinjected (500 nl at 30 ml/min) (Nanoliter2010, WPI) to target the PPT/LDT area ($-4.5$ mm posterior, 1 mm lateral from bregma and 3.25 mm depth from brain surface). The micropipette was left in the brain for an additional 10 min and then slowly raised up. An optic fiber cannula (CFM14L05, Thorlabs) was then implanted 3 mm deep from the surface of the brain and all components were secured to each other and to the skull with dental cement.

## In vivo electrophysiological experiments in a head-fixed condition

Experimental procedures were as described previously (*Lyngholm and Sakata, 2019*; *Yague et al., 2017*). Briefly, all electrophysiological recordings were performed in a single-walled acoustic chamber (MAC-3, IAC Acoustics) with the interior covered with 3 inches (~ 7.5 cm) of acoustic absorption foam. The animals were placed in the same head-fixed apparatus, by securing them by the head-post and placing the animal into an acrylic tube. During the entire recording, the animals were not required to do anything actively. For pontine EEG recording, cortical EEG, EMG and pontine EEG signals were amplified (HST/32 V-G20 and PBX3, Plexon), filtered (0.1 Hz low cut), digitized at a sampling rate of 1 kHz and recorded using LabVIEW software (National Instruments). Recording was performed for 5 hr from 9:00 to 14:00. For brainstem or hippocampal silicon probe recording, a 32-channel four shank silicon probe (A4 $\times$ 8–5 mm-100-400-177 for brainstem recording or Buzsaki32 for hippocampus recording) was inserted slowly with a manual micromanipulator (SM-25A, Narishige) into the brainstem (3.75 mm – 4.3 mm from the brain surface) or the hippocampus (1.55 mm – 1.85 mm from the brain surface). Probes were inserted perpendicularly with respect to the brain surface. Broadband signals were amplified (HST/32 V-G20 and PBX3, Plexon) relative to the screw on the cerebellum, filtered (0.1 Hz low cut), digitized at 20 kHz and recorded using LabVIEW software (National Instruments). The recording session was initiated >1 hr after the probe was inserted to its target depth, to stabilize neural signals. Recording preparation started from 8:00 and terminated by 15:00. For verification of silicon probe tracks, the rear of the probes was painted with DiI (~10% in ethanol, D282, Invitrogen) before insertion.

## Pupil monitoring

In a subset of in vivo electrophysiological experiments under a head-fixed condition, pupils were also monitored with an off-axis infrared (IR) light source (860 nm IR LED, RS Components). A camera (acA1920-25μm, Basler Ace) with a zoom lens (M0814-MP2, Computar) and an IR filter (FGL780, Thorlabs) was placed ~10 cm from the animal's left eye. Images were collected at 25 Hz using a custom-written LabVIEW program and a National Instruments image grabber (PCIe-8242).

## In vivo fiber photometry experiments in a freely behaving condition

The fiber photometry system consisted of two excitation channels. A 470 nm LED (M470L3, Thorlabs) was used to extract a $Ca^{2+}$-dependent signal and a 405 nm LED (M405L3, Thorlabs) was used to obtain a $Ca^{2+}$-independent isosbestic signal. Light from the LEDs was directed through excitation filters (FB470-10, FB405-10, Thorlabs) and a dichroic mirror to the fiber launch (DMLP425R and

KT110/M, respectively). The fiber launch was connected to a multimode patch cable (M82L01, Thorlabs), which attached to an implantable optic fiber on the mouse via a ceramic mating sleeve (CFM14L05 and ADAF1, respectively). Light emissions from GCaMP6s expressing neurons were then collected back through the optic fiber and directed through a detection path, passing a dichroic mirror (MD498) to reach a photodetector (NewFocus 2151, Newport). A National Instruments DAQ (NI USB-6211) and custom-written LabVIEW software were used to control the LEDs and to acquire fluorescence data at 1 kHz. LEDs were alternately turned on and off at 40 Hz in a square pulse pattern. Electrophysiology signals were recorded at 1 KHz using an interface board (RHD2000, Intan Technologies) and connected to the mouse via an amplifier (RHD2132, Intan Technologies). Mice were habituated to being handled and tethered to the freely behaving system over several consecutive days. Mice were scruffed and the straw on the headcap slotted into a custom-made clamp, to keep the head still and to absorb any vertical forces when connecting the electrophysiology and fiber photometry tethers to the headcap. Once connected, mice were placed in an open top Perspex box (21.5 cm x 47 cm x 20 cm depth) lined with absorbent paper, bedding and some baby food. Recordings lasted 4–5 hr to allow for multiple sleep–wake transitions.

## Histological analysis

After electrophysiological experiments, animals were deeply anesthetized with a mixture of pentobarbital and lidocaine, and perfused transcardially with 20 mL saline followed by 20 mL 4% paraformaldehyde/0.1 M phosphate buffer, pH 7.4. The brains were removed and immersed in the above fixative solution overnight at 4°C and then immersed in a 30% sucrose in phosphate buffer saline (PBS) for at least 2 days. The brains were quickly frozen and were cut into coronal or sagittal sections with a sliding microtome (SM2010R, Leica) or with a cryostat (CM3050, Leica), each with a thickness of 50 or 100 μm. The brain sections were incubated with a NeuroTrace 500/525 Green-Fluorescence (1/350, Invitrogen) or NeuroTrace 435/455 Blue-Fluorescence (1/100, Invitrogen) in PBS for 20 min at room temperature (RT), followed by incubation with a blocking solution (10% normal goat serum, NGS, in 0.3% Triton X in PBS, PBST) for 1 hr at RT. For ChAT-IRES-Cre::Ai32 mice, to confirm the position of ChAT-expressing neurons, we performed GFP and ChAT double staining. These brain sections were incubated with mouse anti-GFP antiserum (1/2000, ABCAM) and goat anti-ChAT antiserum (1/1000, Millipore) in 3% NGS in PBST overnight at 4°C. After washing, sections were incubated with DyLight 488-labeled donkey anti-mouse IgG (1/500, Invitrogen) and Alexa 568-labeled donkey anti-goat IgG (1/500, Invitrogen) for 2 hr at RT. After staining, sections were mounted on gelatin-coated or MAS-coated slides and cover-slipped with 50% glycerol in PBS. The sections were examined with a fluorescence microscope (BZ-9000, Keyence).

## Data analysis

### Sleep scoring

Vigilance states were visually scored offline according to standard criteria (*Radulovacki et al., 1984*; *Tobler et al., 1997*). Wakefulness, NREM sleep, or REM sleep state was determined in every 4-s window on the basis of cortical EEG and EMG signals using custom-made MATLAB GUI. For electrophysiological or fiber photometry experiments, the same individual scored all recordings for consistency.

### Pupil analysis

Video files were processed using DeepLabCut (*Mathis et al., 2018*). Each video file consisted of a 5 min segment of the recording, meaning that each experiment yields tens of video files. For training, a single file containing all states (AW, NREM sleep and REM sleep) was chosen on the basis of the sleep score. This initial step was critical to reflect the dynamic range of pupil dilation or constriction in each experiment. After choosing the file, 20 frames were randomly selected to mark the left and right edges of the pupil manually. ImageJ was used for this manual marking. Using these labeled frames, the deep convolutional neural network was then trained, and all of the video files were processed to detect the left and right edges of pupil. After processing, visual inspection was performed by generating a down-sampled (50–100 times) video clip. The same procedure was applied across all recordings. To compute pupil diameter, the distance between the left and right edges of the pupil was calculated across frames. The profile was filtered (low-pass filter at 0.5 Hz) and z-scored.

To compute eye movement, the middle point of pupil was determined and the distance of the middle points between two continuous frames was calculated. The profile was then normalized by the maximal value of the profile. From 20 pupil recordings (14 animals), two recordings from a single animal were excluded because of eye closure during most of recording period (*Supplementary file 1*).

## Fiber photometry signal processing

Custom-written MATLAB scripts were used to compute dF/F signals. To extract 405 and 470 nm signals, illumination periods were determined by detecting synchronization pulses. For each illumination epoch, the median fluorescent signal was calculated. Because each illumination epoch consisted of pulses at 40 Hz, the fluorescent signals originally sampled at 1 kHz were effectively down-sampled to 40 Hz. Photobleaching was estimated by a single exponential curve and the difference between the fluorescent signal trace and the estimate was further low-pass filtered at 4 Hz. To estimate moving artifacts, the filtered 405 nm signals were linearly scaled based on the filtered 470 nm signals using a linear regression. To estimate dF/F signals, then the 470 nm signals were subtracted from the scaled 405 nm signals. In this study, the first 10 min segment was excluded for further analysis because of poor estimation of the photobleaching profile.

## Spike train and LFP/EEG analysis

For spike sorting, Kilosort (*Pachitariu et al., 2016*) was used for automatic processing, followed by manual curation using phy (https://github.com/cortex-lab/phy). Clusters with ≥20 isolation distance were recognized as single units. The position of single units was estimated on the basis of the channel that provided the largest spike amplitude. All subsequent analysis was performed using custom-written codes (MATLAB, Mathworks).

To categorize functional classes of single units, average firing rate for each behavioral state was calculated and a hierarchical clustering approach with the Ward's method was applied. To predict the pupil diameter from single unit activity, spike trains were filtered (band-pass filter between 0.5 and 25 Hz) and then a linear regression analysis was performed. To evaluate the goodness-of-fit of the linear model, the R-squared value was calculated. The same process was repeated by shifting the time relationship between spike trains and pupil diameter to determine an optimal time window to predict pupil diameter from spike train. Then the sequence of R-squared values was normalized by computing Z-scores.

To predict the pupil diameter from simultaneously monitored multiple single unit activities, spike trains were filtered (band-pass filter between 0.5 and 25 Hz) and a linear regression model was trained by using a regularized support-vector machine algorithm with 10-hold cross-validation. Then, cross-validated mean squared error (MSE) was computed. The same process was repeated by shifting the time relationship between spike trains and pupil diameter. The sequence of MSEs were normalized.

To find interpretable patterns in the population activity, we adopted non-negative matrix factorization (NMF) (*Lee and Seung, 1999*; *Onken et al., 2016*). NMF takes a non-negative data matrix and factorizes this matrix into two non-negative matrices. One of these matrices, the module matrix contains patterns, whereas the other matrix, the activation coefficient matrix, indicates how these patterns need to be weighted in order to reconstruct the original data matrix. Owing to the non-negativity constraints, NMF yields a parts-based representation: one pattern cannot be canceled by another pattern. For a data matrix representing images of faces, the method yields face patterns, such as eyes, eyebrows and noses in the module matrix and non-negative weights of these patterns in the activation coefficient matrix (*Lee and Seung, 1999*). For neural population activity, NMF can decompose spike counts into 'space' (neurons) modules and the activation coefficients of these modules (*Onken et al., 2016*). For this purpose, spike trains were discretized by binning them into 4 s intervals, which were equivalent to the time window for sleep scoring (see above). Let $\mathbf{r}(t)$ denote the resulting vector of population spike counts in bin $t$. We represented all population spike count vectors in a matrix $\mathbf{R} = [\mathbf{r}(1)\ \mathbf{r}(2)\ ..\ \mathbf{r}(T)]$ of size $N$ by $T$, where $N$ denotes the number of neurons and $T$ denotes the number of time bins. We then decomposed the matrix $\mathbf{R}$ into two non-negative matrices $\mathbf{W}$ of size $N$ by $m$ and $\mathbf{H}$ of size $m$ by $T$ as follows: $\mathbf{R} = \mathbf{WH}$, where $m$ is the number of modules. To this end, we applied multiplicative update rules (*Lee and Seung, 2001*):

$$H_{ij} \leftarrow H_{ij} \frac{\sum_k W_{ki} R_{kj}/(\mathbf{WH})_{kj}}{\sum_l W_{li}} \qquad W_{ij} \leftarrow W_{ij} \frac{\sum_k H_{jk} R_{ik}/(\mathbf{WH})_{ik}}{\sum_l H_{jl}} \tag{0.0}$$

These update rules minimize the Kullback–Leibler divergence, corresponding to a Poisson noise assumption for the spike counts (*Févotte and Cemgil, 2009*). In each run, we randomly initialized **W** and **H** and applied the update rules up to 100 times or until convergence. For each decomposition, we performed 10 runs and selected the run with the lowest Kullback–Leibler divergence. The *m* columns of **W** then represented the *m* space modules and the *m* corresponding rows of **H** represented their activation coefficients for each time bin.

To select the number of space modules *m*, we evaluated how many modules were needed, so that additional modules did not significantly improve the decomposition. Our procedure was similar to that used in *De Marchis et al. (2013)*. We generated surrogate data by shuffling all of the elements of the matrix **R** and then decomposed the shuffled matrix in the same way as we previously decomposed the original **R**. We quantified the quality of a decomposition using the variance accounted for (VAF) (*Clark et al., 2010*). Starting with one module, we incremented the number of modules until the VAF of the unshuffled data decomposition did not exceed 3/4 of the average VAF of the decompositions of 100 shuffles.

To classify three behavioral states on the basis of the activation coefficients, a linear classifier was trained by fitting a multivariate normal density to each state with 10-fold cross validation. Then classification performance was calculated. This procedure was repeated by shifting the time relationship between the activation coefficients and sleep scores.

To detect P-waves, two EEG or LFP signals in the brainstem were subtracted and filtered (5–30 Hz band-pass filter). If the signals cross a threshold, the event was recognized as P-waves. To determine the detection threshold, a 1 min segment of the subtracted signals was extracted from the longest NREM sleep episode for a reliable estimation of stable noise level. The noise level was estimated by computing root-mean-square (RMS) values in every 10-ms time window. The threshold was defined as mean + 5 x standard deviation of the RMS values. The timing of P-waves was defined as the timing of the negative peak.

The phase analysis was essentially the same as that described previously (*Yague et al., 2017*). Cortical EEG or hippocampal LFP signals were used for this analysis. For hippocampal LFPs, signals from two separate channels were subtracted to minimize volume conduction. To derive band-limited signals in different frequency bands, a Kaiser finite impulse response filter was used with sharp transition bandwidth of 1 Hz, pass-band ripple of 0.01 dB and stop-band attenuation of 50 dB. For filtering, the MATLAB 'filtfilt' function was used. In the present study, the following bands were assessed: [2–4], [4–7], [7–10], [10–15], [15–20], [20–30], [30–50], [50–80], [80–110], and [110–150] Hz. The instantaneous phase of each band was estimated from the Hilbert transform and the phase of P-wave occurrence was computed. To quantify the relationship between P-waves and EEG/LFP phase, the percentage of P-waves elicited in each phase bin was calculated. The resultant vector length was computed as an absolute value of mean resultant vector $\bar{r}$ by using the *circ_r* function of the CircStats Toolbox (*Berens, 2009*). Rayleigh's test for non-uniformity of circular data was performed to assess the statistical significance ($p < 0.01$) of the non-uniformity of the P-wave vs EEG/LFP phase distribution.

To detect ripples in the hippocampus, LFP signals from the channel that detected spiking activity were used. Band-limited signals at 140–250 Hz were computed by using a Kaiser finite impulse response filter (see above). Two sequences of RMS values were calculated with two different time window: 2 s (long RMS) and 8 ms (short RMS). If the short RMS exceeds four times larger long RMS for 8 ms, then signals were recognized as a ripple event. To quantify the asymmetry of the cross-correlogram between SWRs and P-waves (*Figure 7D*), the asymmetry index was defined as: ($N_{pre}$ – $N_{post}$)/($N_{pre}$ + $N_{post}$), where $N_{pre}$ and $N_{post}$ are the number of SWRs 100 ms before and after P-waves, respectively.

## Statistical analysis

Data were presented as mean ± SEM unless otherwise stated. Statistical analyses were performed with MATLAB. In *Figures 2G* and *3A*, one-way ANOVA with post-hoc Tukey's Honest Significant Difference (HSD) criterion was performed. In *Figure 4C*, repeated measures ANOVA was performed. In

*Figure 5C, a* two-tailed *t*-test was performed. In *Figure 6C*, Rayleigh test for non-uniformity was performed. In *Figures 6B* and *7B*, two-way ANOVA with post-hoc HSD criterion was performed. In *Figure 7D, a* two-tailed *t*-test was performed. To estimate effect size, Hedges' *g* was computed using the Measures of Effect Size Toolbox (*Hentschke and Stüttgen, 2011*).

## Code availability

The code is available on GitHub (https://github.com/Sakata-Lab; copies archived at https://github.com/elifesciences-publications).

## Acknowledgements

This work was supported by the BBSRC (BB/M00905X/1), the Leverhulme Trust (RPG-2015–377), Alzheimer's Research UK (ARUK-PPG2017B-005), and Action on Hearing Loss (S45) to SS, by the EPSRC (EP/S005692/1) to AO, and by a JSPS Postdoctoral Fellowship for Research Abroad, a Research Fellowship from the Uehara Memorial Foundation, and PRESTO from JST and KAKENHI (17H06520) to TT.

## Additional information

### Funding

| Funder | Grant reference number | Author |
|---|---|---|
| Biotechnology and Biological Sciences Research Council | BB/M00905X/1 | Shuzo Sakata |
| Leverhulme Trust | RPG-2015-377 | Shuzo Sakata |
| Alzheimer's Research UK | ARUK-PPG2017B-005 | Shuzo Sakata |
| Action on Hearing Loss | S45 | Shuzo Sakata |
| Engineering and Physical Sciences Research Council | EP/S005692/1 | Arno Onken |
| Japan Society for the Promotion of Science | | Tomomi Tsunematsu |
| Uehara Memorial Foundation | | Tomomi Tsunematsu |
| Japan Science and Technology Agency | JPMJPR1887 | Tomomi Tsunematsu |
| Japan Society for the Promotion of Science | 17H06520 | Tomomi Tsunematsu |

The funders had no role in study design, data collection and interpretation, or the decision to submit the work for publication.

### Author contributions

Tomomi Tsunematsu, Data curation, Formal analysis, Funding acquisition, Investigation, Methodology; Amisha A Patel, Data curation, Formal analysis, Investigation, Methodology; Arno Onken, Software, Funding acquisition, Methodology; Shuzo Sakata, Conceptualization, Resources, Data curation, Software, Formal analysis, Supervision, Funding acquisition, Validation, Investigation, Visualization, Methodology, Project administration

### Author ORCIDs

Arno Onken (iD) http://orcid.org/0000-0001-7387-5535
Shuzo Sakata (iD) https://orcid.org/0000-0001-6796-411X

### Ethics

Animal experimentation: All experimental procedures were performed in accordance with the United Kingdom Animals (Scientific Procedures) Act of 1986 Home Office regulations and approved by the Home Office (PPL 70/8883).

Decision letter and Author response
Decision letter https://doi.org/10.7554/eLife.52244.sa1
Author response https://doi.org/10.7554/eLife.52244.sa2

## Additional files

### Supplementary files

• Supplementary file 1. Summary of experimental animals and recordings involved in this study. BS-Si, silicon probe recording in the brainstem. HP-Si, silicon probe recording in the hippocampus. BS-EEG, EEG recording in the brainstem. FP, fiber photometry. Exc[1], the animal was excluded because of electrode mispositioning. Exc[2], the animal was excluded because of lack of histological data. Exc[3], the animal was excluded because of eye closure during recording.

• Transparent reporting form

### Data availability

All sorted spike trains, sleep scores, fluorescent signals, P-wave timing and pupil information are available online at https://doi.org/10.15129/4f81777a-7b88-48f3-af4d-da484706fa5d.

The following dataset was generated:

| Author(s) | Year | Dataset title | Dataset URL | Database and Identifier |
|---|---|---|---|---|
| Tsunematsu T, Patel AA, Onken A, Sakata S | 2019 | Data for: State-dependent brainstem ensemble dynamics and their interactions with hippocampus across sleep states | https://doi.org/10.15129/4f81777a-7b88-48f3-af4d-da484706fa5d | University of Strathclyde, 10.15129/4f81777a-7b88-48f3-af4d-da484706fa5d |

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
