## [Decision Letter]

**Acceptance summary:**

The results of this manuscript, for the first time, provide experimental evidence to shed light on a long-standing question of how pontine-wave (P-wave) activity interacts with hippocampal network activity patterns during non-REM and REM sleep. Over the last 2 decades, groups of investigators working on hippocampal physiology have argued that SWRs are the primary physiological mechanism underlying sleep-associated memory consolidation. Yet, another group of researchers have argued that the P-wave is the primary physiological activity for sleep-associated memory consolidation. This electrophysiological study correlates pontine neuronal ensemble dynamics during P-waves with hippocampal SWRs during non-REM sleep. Thus, the results of the present study reveal an important link between P-waves and SWRs. This study also shows, for the first time, that the P-waves are present in mice. The characterization of P-waves in mice is expected to pave the way for future studies of specific P-wave mechanisms of generation as well as impacting a broad range of studies that address state-specific brain stem influences on neuronal ensemble dynamics in higher brain regions.

**Decision letter after peer review:**

Thank you for submitting your article "State-dependent pontine ensemble dynamics and interactions with cortex across sleep states" for consideration by *eLife*. Your article has been reviewed by three peer reviewers, and the evaluation has been overseen by Laura Colgin as the Senior Editor. The following individuals involved in review of your submission have agreed to reveal their identity: Subimal Datta (Reviewer #3).

The reviewers have discussed the reviews with one another and the Reviewing Editor has drafted this decision to help you prepare a revised submission.

Essential revisions:

1) The current title of the manuscript is 'State-dependent pontine ensembles dynamics.…', and the Abstract and main text refer to "pontine populations" and "pontine neurons". However, for the majority of analysis, a mixture of brainstem neurons including neurons in the midbrain and medulla were used. If the authors would like to uncover effects that are specific to pons, they should use only neurons recorded from pons. In that case, detailed histological verification would be required for each animal. A histological analysis appears in Figure 1—figure supplement 1, but it is only provided for one animal and with low resolution; therefore, it is impossible to determine from this image from which brain stem structure the neurons were recorded. Relatedly, Figure 1—figure supplement 1 refers to "Neurons recorded from pontine cholinergic nuclei", yet histological analyses verifying that these neurons were recorded from pontine cholinergic nuclei are missing.

2) The relationship of the present work to previous work requires further elaboration. Specific points related to this point are listed below:

a) In the cat, PGO-waves have been recorded in the lateral geniculate nucleus and in many other parts of the forebrain. The pons has been suspected to be the origin of the first component of the PGO waves in the forebrain and occipital cortex. However, in the rat, the PGO-wave is absent in the lateral geniculate and other forebrain areas. Therefore, the functions of P-waves in the rats and mice may not be similar as the functions of PGO waves in the cat. This possibility should be acknowledged in this manuscript. The P-wave generator in the rat (1998) and P-wave/PGO-wave in generator in the cat (1992) were shown to be in two different areas in the pons. This study, for the first time, shows that the P-waves are present in mice and will likely encourage future investigations to uncover mechanisms of P-wave generation in mice, which may be different than cats and rats.

b) The anticipatory relationship between neuronal activity in the pons and state changes has been shown previously by Steriade, McCarley, Hobson, Datta and colleagues (reviewed in the classic book, "Brain control of wakefulness and sleep" and other review articles by these authors. See also Steriade, Sakai and Jouvet 1984). Karashima and colleagues have described the relationship between pontine activity and hippocampal theta in cats (Brain Res 2002 958: 347-358, not cited) and rats (Brain Res 2005 1051: 50-6, not cited).

c) Relevant studies involving single-unit recordings that described PGO burst neurons (e.g. McCarley, Nelson and Hobson, 1978) should be discussed.

3) The authors recorded neuronal activity during sleep from head-fixed mice, whose eyes were open during sleep session. A point was raised that mice naturally sleep with their eyes closed. Also, the structure and properties of sleep are likely different between freely moving (non-fixed) and head-fixed animals. Free-moving mice show NREM-REM-NREM cycles in addition to the NREM-REM-AWAKE sequence. The authors only showed the NREM-REM-AWAKE transition and not a NREM-REM-NREM transition. The latter transition may not occur because sleep in head-fixed animals differs from that of free-moving animals. Therefore, the authors should compare and contrast the properties (e.g., duration, EMG, LFP of REM and non-REM, cycles and structure of NREM-REM-NREM) of sleep in head-fixed animals with those properties that have been reported in freely moving animals.

4) It is unclear whether the differences between REM and NREM in Figure 6D, E and Figure 7B are statistically significant.

5) It is unclear from which area (CA1, CA3, DG) of the (dorsal or ventral) hippocampus recordings were obtained.

6) The authors should clarify what the animals were doing during the awake state.

---

## [Author Response]

Essential revisions:1) The current title of the manuscript is 'State-dependent pontine ensembles dynamics.…', and the Abstract and main text refer to "pontine populations" and "pontine neurons". However, for the majority of analysis, a mixture of brainstem neurons including neurons in the midbrain and medulla were used. If the authors would like to uncover effects that are specific to pons, they should use only neurons recorded from pons. In that case, detailed histological verification would be required for each animal. A histological analysis appears in Figure 1—figure supplement 1, but it is only provided for one animal and with low resolution; therefore, it is impossible to determine from this image from which brain stem structure the neurons were recorded. Relatedly, Figure 1—figure supplement 1 refers to "Neurons recorded from pontine cholinergic nuclei", yet histological analyses verifying that these neurons were recorded from pontine cholinergic nuclei are missing.

We agree that our original title and some descriptions in the main text were misleading. It was not our intention to claim that we have uncovered the effects specific to the pons.

To correct this, we have made the following two amendments.

First, we have amended the title as follows: “State-dependent *brainstem* ensemble dynamics and their interactions with *hippocampus* across sleep states”.

Second, we have also edited the Abstract and main text, especially the Introduction. We have replaced “pons” by brainstem where appropriate. Please check the highlighted parts in the Abstract, Introduction, Discussion and Materials and methods.

In addition, to address the concern about Figure 2—figure supplement 1, we have revised the figure thoroughly by adding a histological image as well as a single unit activity recorded from the LDT.

2) The relationship of the present work to previous work requires further elaboration. Specific points related to this point are listed below:a) In the cat, PGO-waves have been recorded in the lateral geniculate nucleus and in many other parts of the forebrain. The pons has been suspected to be the origin of the first component of the PGO waves in the forebrain and occipital cortex. However, in the rat, the PGO-wave is absent in the lateral geniculate and other forebrain areas. Therefore, the functions of P-waves in the rats and mice may not be similar as the functions of PGO waves in the cat. This possibility should be acknowledged in this manuscript. The P-wave generator in the rat (1998) and P-wave/PGO-wave in generator in the cat (1992) were shown to be in two different areas in the pons. This study, for the first time, shows that the P-waves are present in mice and will likely encourage future investigations to uncover mechanisms of P-wave generation in mice, which may be different than cats and rats.

Thank you for pointing out the important issue regarding species differences in the mechanism of P/PGO wave generation and their functional implications. We have added comments in Discussion by referring to the pioneering works in the cat and rat.

Discussion section:

“Datta and his colleagues performed a series of pioneering studies and reported species differences in the site of P-wave genesis between cats and rats (Datta et al., 1998, Datta and Hobson, 1995, Datta et al., 1992): the caudolateral peribrachial area was identified as the induction site of P-waves (PGO waves) in cats (Datta and Hobson, 1995, Datta et al., 1992) whereas the subcoeruleus nucleus was identified in rats (Datta et al., 1998). It would be important to revisit this issue in mice by adopting modern technologies.”

Discussion section:

“This subtle difference may be explained by anatomical differences between species (Datta, 2012). These species differences may also imply differences in the function of P-waves between species.”

b) The anticipatory relationship between neuronal activity in the pons and state changes has been shown previously by Steriade, McCarley, Hobson, Datta and colleagues (reviewed in the classic book, "Brain control of wakefulness and sleep" and other review articles by these authors. See also Steriade, Sakai and Jouvet, 1984).

Thank you for pointing out these pioneering works. In the revised manuscript, we have cited the suggested works. In addition, because the present study has utilized population activity to predict vigilance states, we have added a phrase in the Discussion.

Discussion section:

“State-dependent activity in the brainstem has been described over the past several decades by using various methods (Pace-Schott and Hobson, 2002, Datta and Maclean, 2007, Steriade and McCarley, 1990).”

Discussion section:

“Our results in Figures 3 and 4 are consistent with the notion that brainstem populations play a regulatory role in pupil dilation/constriction (Aston-Jones and Cohen, 2005, Larsen and Waters, 2018) as well as global brain states (Brown et al., 2012, Herice et al., 2019, Luppi et al., 2012, Steriade et al., 1984, Weber and Dan, 2016).”

Discussion section:

“This approach allowed us (1) to quantify state dependency of brainstem neural ensemble dynamics on a timescale of seconds to minutes, (2) to decode vigilance states based on population activity, and (3) to characterize neural population activity underlying P-waves for the first time.”

Karashima and colleagues have described the relationship between pontine activity and hippocampal theta in cats (Brain Res 2002 958: 347-358, not cited) and rats (Brain Res 2005 1051: 50-6, not cited).

We have cited their publications.

Discussion section:

“The temporal correlation between P-waves and hippocampal theta rhythms during REM sleep is consistent with previous studies in cats and rats (Karashima et al., 2002, Karashima et al., 2004, Karashima et al., 2005, Sakai et al., 1973).”

c) Relevant studies involving single-unit recordings that described PGO burst neurons (e.g. McCarley, Nelson and Hobson, 1978) should be discussed.

Again, thank you for pointing out this crucial point. We have added comments in Discussion by referring to several relevant works.

Discussion section:

“Indeed, the phasic firing just before (<25 ms) the trough of P-waves (Figure 5) resembles the observations in the cat (McCarley et al., 1978, Steriade et al., 1990, Sakai and Jouvet, 1980, Nelson et al., 1983). In the near future, it would be interesting to extend our approach further to explore how activity propagates across brainstem neurons during P-waves by identifying cell-types.”

3) The authors recorded neuronal activity during sleep from head-fixed mice, whose eyes were open during sleep session. A point was raised that mice naturally sleep with their eyes closed. Also, the structure and properties of sleep are likely different between freely moving (non-fixed) and head-fixed animals. Free-moving mice show NREM-REM-NREM cycles in addition to the NREM-REM-AWAKE sequence. The authors only showed the NREM-REM-AWAKE transition and not a NREM-REM-NREM transition. The latter tranistion may not occur because sleep in head-fixed animals differs from that of free-moving animals. Therefore, the authors should compare and contrast the properties (e.g., duration, EMG, LFP of REM and non-REM, cycles and structure of NREM-REM-NREM) of sleep in head-fixed animals with those properties that have been reported in freely moving animals.

Thank you for raising the important point. Fortunately, our fiber photometry experiments were performed in a tethered condition (hereafter called an “unfixed” condition). Therefore, we had an opportunity to directly compare their sleep architecture, cortical EEGs and EMGs between the head-fixed (n = 24) and unfixed conditions (n = 12). The results have been provided as new Figure 1—figure supplement 3.

Although the total recording duration was significantly different between the conditions (Figure 1—figure supplement 3A), we have observed that the proportion of awake state was significantly longer in our head-fixed condition (Figure 1—figure supplement 3B). This observation was paralleled with the longer average duration of awake episodes (Figure 1—figure supplement 3C). These results may have been reflecting the nature of the head-fixed condition.

Regarding the NREM-REM-NREM transition, we observed only one incidence in our dataset (Figure 1—figure supplement 3D). According to literature (e.g., Figure 3 in Mochizuki et al., *J Neurosci* 2004), the REM-NREM transition is indeed very rare in mice. Because of the extremely low probability, we could not analyse such a state transition.

Please also see Figure 1—figure supplement 3D, in which we did not observe any significant difference in state transition probabilities between the conditions.

In addition to sleep architecture, we have also compared EEGs and EMGs (Figure 1—figure supplement 3E and F). We observed that EMG power (based on root-mean-square) during wakefulness was significantly larger in the unfixed condition. This may be due to the nature of the condition where the animals could move.

Overall, although we have observed several differences in sleep architecture and EMG, we did not observe any significant difference in transition probabilities and cortical EEGs.

4) It is unclear whether the differences between REM and NREM in Figure 6D, E and Figure 7B are statistically significant.

Thank you for pointing out the issue with lacking statistical analysis in the original manuscript. In the revised manuscript, we have provided the outcomes of two-way ANOVA to assess the effect of brain states on the strength of coupling between P-waves and various oscillations. The effect size has provided an additional measure to quantify the effect of states on the coupling at each frequency band.

During this assessment, we have learned that resultant vector strength, a descriptive circular statistic, provides more straightforward interpretation regarding the coupling between oscillations and P-waves. Therefore, we have decided to take this measure, rather than the modulation index.

The results of statistical analysis can be summarised as follows:

– In Figure 6D, the effect of sleep states on the resultant vector strength was significant (F_(1,179)_ = 4.90, p < 0.05, two-way ANOVA).

– In Figure 7B, the effect of sleep states on the resultant vector strength was significant (F_(1,79)_ = 4.00, p < 0.05, two-way ANOVA).

Results section:

“We further assessed this phase-locking activity by computing resultant vector length (Figure 6D), which is a descriptive circular statistic and represents the length of the mean resultant vector (Berens, 2009): the closer this value is to one, the more concentrated is the phase coupling.”

Results section:

“We found that the effect of sleep states on these phase-locking activity was statistically significant (F1,179 = 4.90, *p* < 0.05, two-way ANOVA). We also computed effect size (Figure 6E). The effect was larger at the delta range during NREM sleep. Hence, the temporal coupling between P-waves and cortical oscillations was modified by sleep states.”

Results section:

“We found that the effect of sleep states on the resultant vector length was significant (F1,79 = 4.00, *p* < 0.05, two-way ANOVA). While the timing of P-waves was strongly phase-locked at the theta range (~7 Hz) in both sleep states and there was no significant difference in the resultant vector length across frequencies, we observed stronger phase modulations in high frequency (≥30 Hz) components during REM sleep based on the effect size.”

Materials and methods section:

“The resultant vector length was computed as an absolute value of mean resultant vector 𝑟̅ by using *circ_r* function of CircStats Toolbox (Berens, 2009).”

Materials and methods section:

“In Figures 6B and 7B, two-way ANOVA with post-hocHSD criterion was performed. In Figure 7D, two-tailed *t*-test was performed.”

In addition, we have also noticed that we did not provide evidence how the results of Figure 7D (the coupling between P-waves and SWRs during NREM sleep) are replicable.

In the revised manuscript, we have addressed this issue by quantifying the asymmetry of the cross-correlogram by measuring the “asymmetry index”. We have found that the number of SWRs was higher before P-waves than that after P-waves across all recordings (*n* = 4) (*p* < 0.05, *t*-test), indicating that the temporal relationship between P-waves and SWRs was robust.

Please see Results and Materials and methods, and the inset of Figure 7D.

Results section:

“We found that ripple events preceded P-waves during NREM sleep across all of four recordings (Figure 7D inset) (*p* < 0.05, *t*-test).”

Materials and methods section:

“To quantify the asymmetry of the cross-correlogram between SWRs and P-waves (Figure 7D), the asymmetry index was defined as: (*N_pre_- N_post_)/(N_pre_*+ *N_post_*), where *N_pre_*and *N_post_*are the number of SWRs 100 ms before and after P-waves, respectively.”

5) It is unclear from which area (CA1, CA3, DG) of the (dorsal or ventral) hippocampus recordings were obtained.

The hippocampus recordings were performed in the dorsal CA1. We have added a new figure as Figure 4 —figure supplement 2.

6) The authors should clarify what the animals were doing during the awake state.

The animals were not required to do anything actively during the entire recording. To clarify this, we have added sentences in the Materials and methods section.

Materials and methods section:

“The animals were placed in the same head-fixed apparatus, by securing them by the head-post and placing the animal into an acrylic tube. During the entire recording, the animals were not required to do anything actively.”